# Where is your field going? A machine learning approach to study the relative motion of the domains of physics

**Andrea Palmucci[1], Hao Liao[2], Andrea Napoletano** [1,3]*, **Andrea Zaccaria[3]**

**1** Department of Physics, "Sapienza" University of Rome, Rome, Italy, **2** Guangdong Province Key Laboratory of Popular High Performance Computers, College of Computer Science and Software Engineering, Shenzhen University, Shenzhen, China, **3** Institute for Complex Systems, UOS Sapienza, Rome, Italy

\* andrea.napoletano1990@gmail.com

## Abstract

We propose an original approach to describe the scientific progress in a quantitative way. Using innovative Machine Learning techniques we create a vector representation for the PACS codes and we use them to represent the relative movements of the various domains of Physics in a multi-dimensional space. This methodology unveils about 25 years of scientific trends, enables us to predict innovative couplings of fields, and illustrates how Nobel Prize papers and APS milestones drive the future convergence of previously unrelated fields.

## 1 Introduction

We aim at building a quantitative framework to describe the time evolution of scientific fields and to make predictions about their relative dynamics. Scientific progress [1] has been already investigated from multiple points of view [2], that range from the study of scientific careers and the evolution of single scientific fields to the mutual impacts between science and society. This latter issue is greatly influenced by the availability of prediction models. For instance, Martinez et al. investigate the impact on education and labour of technological and scientific progress and on the feedbacks which in turn are given from education and labour market to science and technology [3]. Börner et al., instead, discuss the importance of having reliable predictive models in science, technology and economics paired with an easily readable data visualization procedure to help policy makers in their activity [4]. As we will show in the following, our methodology allows for concrete predictions about the time evolution of scientific fields.

Another successfull field of research investigates the scientific careers. Shneiderman discusses in details the best strategies for producing highly successful scientific researches balancing between the exploration of new ideas and the exploitation of established works [5]. Ma et al. [6] and Sinatra et al. [7] both focus on the individual impact of scientists, the former by analyzing the collaboration network of scientific-prize-winners, the latter focusing on the evaluation of the activity of scientists. Jia et al. [8] have introduced a random walk based model to

**Data Availability Statement:** Access to the original APS Data Sets for Research can be requested from the American Physical Society (https://journals.aps.org/datasets). Processed data and codes can are

available through GitHub (https://github.com/Andrea-Napoletano/WyFiG).

**Funding:** The authors acknowledge support from CNR Progetto di Interesse CRISIS LAB, the National Natural Science Foundation of China (Grant Nos. 61803266, 61703281, 91846301, 71790615), Guangdong Province Natural Science Foundation (Grant Nos. 2019A1515011173, 2019A1515011064, 2017B030314073), Shenzhen Fundamental Research-general project (JCYJ20190808162601658).

**Competing interests:** NO authors have competing interests.

investigate the interest change in scientific careers and how they evolve together with the scientific progress. All these studies could benefit from a comprehensive representation of the space in which such careers take place. Indeed, others scientists have contributed to shed light on some of the fundamental mechanisms and underlying rules of the scientific progress: which are the successful strategies to conduct a scientific project, how much the scientific progress is shaped by citations and collaborations networks, see [9–14]. In this respect, a key element is to be able to efficiently project the dynamics of science in a suitable *space*, to obtain both a visualization and, if possible, a prediction of what will happen in the future. Many authors have tackled this issue employing the instruments provided by network theory. Gerlach et al. [15] for example developed an innovative topic model that exploits community detection techniques, Herrera et al. [16] focused on building a network of PACS that they use to study the established communities of fields and their evolution, Sun et al. [17] adopted a network based approach which exploits co-occurrences of authors, Pugliese et al. grounded their analysis on the co-occurrences of sectors in countries [18]. In a recent paper, Chinazzi et al. [19], intruduce a knowledge map of PACS produced by a general-purpose embedding algorithm, StarSpace [20]. They rely on the publication patterns of authors to define a metric of similarity between PACS, and use it to analyze the spatial distribution over different cities of the scientific activity and how it relates with the standard socioeconomic indicators provided by World Bank.

Here we propose a framework which is, instead, well suited to highlight the dynamic of scientific progress. In our analysis, in a way similar to [19], we move from traditional topological spaces, such as networks of PACS or authors, to a continuous space where it is possible to introduce quantitative measures of proximity between scientific topics and most importantly, track their evolution through time. In particular, we represent PACS as multidimensional vectors, leveraging on the methodology discussed in [21] and on Natural Language Processing techniques [22, 23]. The key idea is to draw a parallel between PACS and words, i.e. PACS are the words of what we call *scientific language*. A direct consequence of this novel way to look at PACS is to consider scientific articles as sentences, i.e. contexts which subsume the underlying rules of the *scientific language* as much as a sentence subsumes the underlying syntactic rules of the natural language in which it is formulated. This assumption allows to create a similarity metric between scientific fields, that we call *context similarity*. While in [21] this approach was introduced and used to forecast new combinations of the technological codes to make prediction on the future patenting activity, here we aim to quantitatively measure scientific trends in the Physics literature by looking at the dynamics PACS codes. This enables us not only to predict new combinations of fields but also to assess the impact of extra-ordinary contributions such as Nobel Prize papers and APS milestones.

The rest of this paper is organized as follows. We first show how the mere representation of PACS dynamics in a low dimensional space gives a series of insights about how research in Physics clusterize and how scientific fields move one with respect to the others. Then we use these relative movements to forecast the appearance of innovative couplings. We also show that the publication of recognized papers is followed by an approach of the relative PACS. In the last section, we discuss in more detail the database and the methodology we used to build our representation of PACS from the data.

## 2 Results

### 2.1 Low dimensional representation

The vector representations of PACS, which we call *embeddings*, live in a high dimensional space, and this prevents a direct inspection of the resulting structures. In the Methods Section we provide more details on the algorithm that constructs them staring from the raw data. For

the purpose of understanding the results presented here, it suffices to know that the position of these high-dimensional vectors in the space of PACS is optimized so that each of them has as neighbours the most similar ones given the global scientific activity (the concept of similarity is quantified through the scalar product between vectors, see Methods for more details). A simple visualization of these representations and their time evolution is required to shed light on the dynamics underlying the scientific activity in Physics. For this reason we rely on a standard dimensionality reduction technique that allows us to generate a two dimensional representations of our embeddings. We use the t-SNE algorithm (t-distributed Stochastic Neighbor Embedding) [24] and its modification that takes into account time-ordered input data, Dynamic t-SNE [25]. Dynamic t-SNE requires the different instances of the high dimensional space to contain the same embeddings, because it keeps track of them to preserve temporal coherence between consecutive projections. In this way, the 2-dimensional projection at time $t + 1$, not only depends on the high dimensional configuration of the embeddings at the same time, but also on the 2-dimensional projection at time $t$. In particular, the projections at time $t$ are used as the initial conditions for projections at time $t + 1$ in order to reconstruct coherent trajectories. For this reason, we have restricted the number of PACS by selecting only those present into the whole time range under investigation, i.e. 1985-2009, for a total of about 300 PACS.

The result of the dimensional reduction is shown in Fig 1 where we have added the ellipses to stress the cluster structure. As expected, most of the PACS are clusterized respecting the hierarchy of the classification (see the Data and methods section), that is represented by the different colors of the PACS trajectories. The relative position of the clusters is in very good agreement with intuition: Nuclear Physics is close to Elementary Particles and fields, the two Condensed Matter clusters are also close, while the General and interdisciplinary sectors are not clearly localized. In some interesting cases some PACS are not localized into their original cluster coming from the PACS classification. We name some of these noteworthy exceptions:

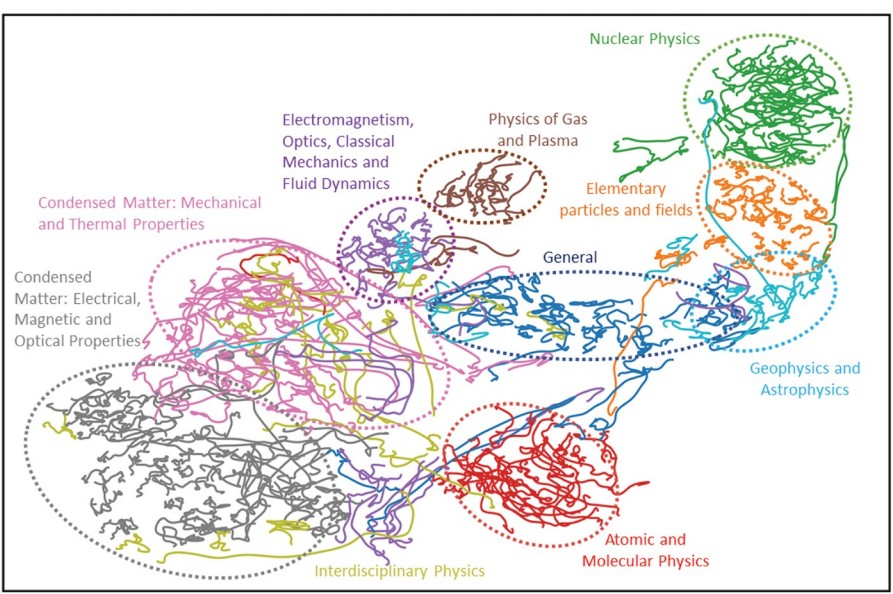

**Fig 1. Two dimensional representation of the PACS embeddings.** The dynamical evolution of the scientific domains of Physics follows only partially the hierarchical classification. See the text for details.

- **Quantum Electrodynamics(Orange)** Its trajectory starts in 1985 from its Elementary Particle Fields cluster (Orange) and arrives in 2009 in the Atomic and Molecular Physics cluster (Red).

- **Stellar Characteristics and Properties (Sky Blue)** It starts from the Nuclear Physics cluster (Green) and arrives in its cluster Geophysics and Astrophysics (Sky Blue).

- **Properties and Dynamics of the Atmosphere, Meteorology (Sky Blue) and Physical Oceanography 92.10 (Sky Blue)** Both fields can be found within the Electromagnetism, Optics, Classical Mechanics and Fluid Dynamics cluster (Violet).

- **Macro-molecules and Polymer Molecules (Red)** It moves inside the Condensed Matter cluster (Purple)

- **Physical Properties of Rocks and Minerals (Sky Blue)** It is inside the cluster Condensed Matter: Mechanical and Thermal Properties (Purple).

All the previous examples show PACS whose use and dynamics does not reflect their classification.

We believe that this representation can have a number of practical applications. For instance, it could be used to update and redesign the classification of research domains and to improve the synergies among researches of (supposedly) different areas.

## 2.2 Prediction of new PACS pairs

*Context similarity* is a metric introduced in [21] which we have specifically adapted to measure the proximity of two PACS given the current scientific production: it mirrors and summarizes the relationship between their respective scientific areas in a given time window. It is therefore natural to use it to estimate the likelihood that a pair of PACS, which has never appeared in a paper before, will occur in the same paper in the future. In our opinion this kind of events can be regarded as an *innovation* in the field of Physics: following the seminal ideas of B.W. Arthur, an innovation is defined as a previously unseen combination of existing elements [26]. In this section we make systematic predictions for the appearance of new PACS pairs and we confirm the goodness of our approach using both the Receiver Operating Characteristic curve (ROC) and its integral (AUC), and the best F1-score, both of them standard tools in statistical analysis [28–30]. As discussed in the Methods sections, scientific articles are grouped in 5-years-long training sets. In order to test the predictive power of *context similarity* we repeat our analysis on 10-years-long time windows formed by joining together two consequent non-overlapping time intervals, e.g. 1985-1989 for training and 1990-1994 for testing. The idea is to test whether the *context similarity* of PACS couples is connected to the likelihood that a previously unseen couple will appear in the testing set. In each 10-years window, we proceed as follows:

1. We use the training set to calculate the embeddings for the 500 most frequent PACS and identify all couples that have never been published together up to the last year of the training set.

2. We check whether couples of PACS selected in the training set are published in at least one paper of the testing set or not. We classify the testing set couples of PACS in two separate classes accordingly: class 0 if they appear together in at least one paper, class 1 otherwise.

3. We evaluate the effectiveness of *context similarity* to forecast unseen PACS couples using standard performance metrics such as the ROC-AUC and the best F1-score.

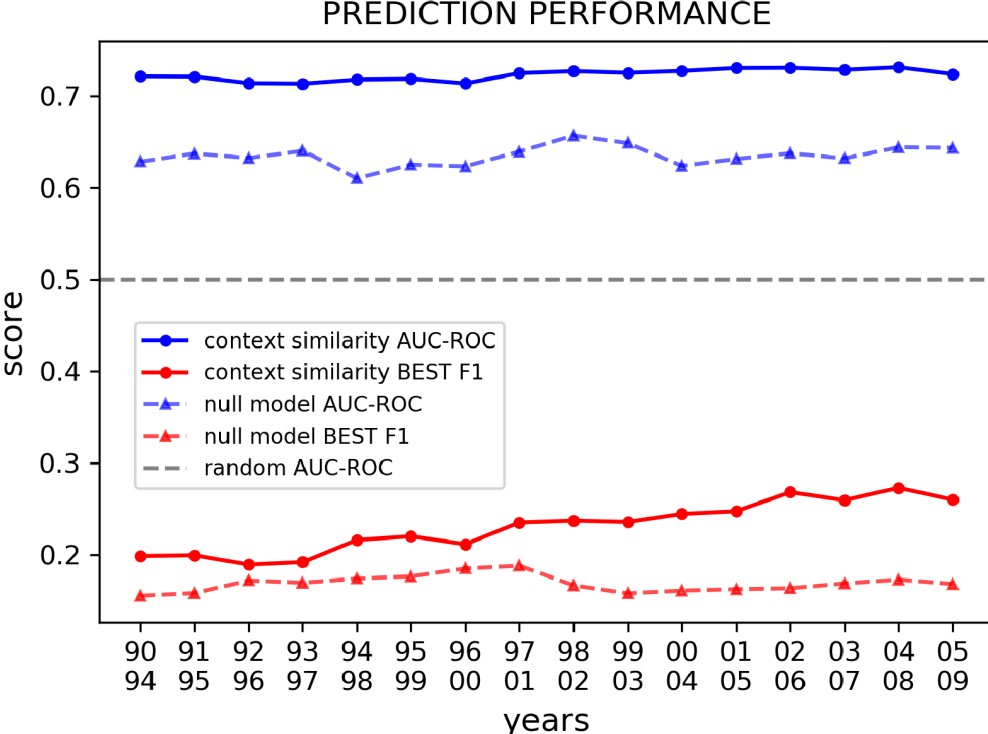

**Fig 2.** *Context similarity* **(continuous lines) outperforms the null model (dashed lines) in predicting innovations in Physics, i.e. new pairs of PACS used for the first time in a paper.** The two metrics are evaluated by the ROC AUC (blue lines) and the Best F1 score (red lines), and the plot show that *context similarity* scores higher in both cases. The database is organized in 10-years-long sliding windows, the first five years of each window form the training set where we calculate the two metrics, while the second five years form the testing set where we evaluate them. On the x axis we report the first and the last year of each testing set. The null model proposed in this plot assign to each paper a random set of PACS in such a way that each article has the same number of PACS and each PACS appears in the same number of articles. The dashed grey line represent the ROC AUC of a random guess.

We test our metric against a null model that takes into account the relative growth of each field relative to the others. To realize this null model we make use of the *curveball* algorithm introduced in [27] to create randomized bipartite networks that preserve the number of connections of each node. In other words, each article gets a random set of PACS in such a way that the following conditions are true:

- All articles in the randomized articles-PACS network have the same number of PACS as they have in the original bipartite network.

- All PACS in the randomized articles-PACS network appear in the same number of articles as they do in the original bipartite network.

We then calculate the embeddings and the *context similarity* over the randomized database. In this way we can test the prediction power of *context similarity* against a null model that, on one side, keeps tracks of the growth of each scientific field through the frequency with which PACS are employed, while on the other, it randomizes all information about the semantic relations between PACS. Due to the long computation time required to calculate the *context similarity* between PACS of the randomized articles-PACS network in every decade, which we refer to as null model for simplicity, we have done it only once for the whole dataset. Results are shown in Fig 2 where we display the ROC AUC and the best F1 score. Regarding the AUC

metric, *context similarity* outperforms the null model proposed and scores well above a random classifier. Indeed, it can be proven that a random classifier would be characterized by a AUC score of 0.5 regardless of the class imbalance ratio of the system under observation [28–30]. Regarding the best F1 score, it can not be directly compared with a random guess, being an harmonic mean of the Precision and Recall mesures [30], but we can compare it against the null model that we have built, and it is evident that *context similarity* still performs better. In summary, *context similarity* captures more information than what can be understood looking just at the frequency of PACS, and therefore, not only it successfully grasps the relation between PACS induced by the global scientific activity, but it is also able to predict innovations in the field of Physics over the years with a constant good performance.

## 2.3 Quantifying the impact of milestones and Nobel prize winners

In Fig 3, we have focused our attention on one illustrative example of PACS dynamics influencing and being influenced by scientific papers to show the effectiveness of the proposed framework to study the evolution of the relation between different fields of research. Highlighted in blue, there are the trajectories of two PACS: *Matter Waves* and *Quantum Statistical Mechanics*: these are the PACS of the Nobel prize article on the Bose-Einstein condensation [31], published in 1995. The publication of this fundamental paper is associated, in the plot, to its PACS converging towards a closer position.

In the following we will study more examples of the effect of both APS milestones (see description of the data for the definition of milestones, and Nobel prize winners on the space of PACS). Each PACS is added to papers by the authors at the time of submission. Under the reasonable assumption that authors follow the order of relevance of the topics, we consider only the first two PACS, i.e. the two main topics of a paper. In total there are 36 of such special articles, however only 20 of them have the first two PACS different at our level of aggregation (4 digits): we calculate the value of *context similarity* for each of them. The aim is to compare the relative variations in *context similarity* of these pairs with the average variations of all

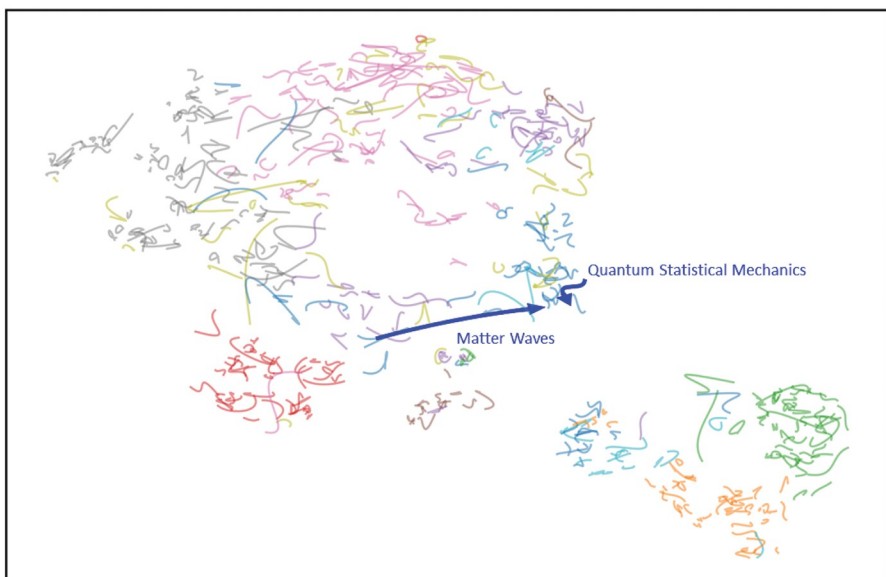

**Fig 3. Dynamics of the first two PACS of the Davis et al. [31] paper, that led to a Nobel prize in 2001 during the time interval 1990-1999.** It is evident how the Matter Waves field moves towards the cluster that contains Quantum Statistical Mechanics.

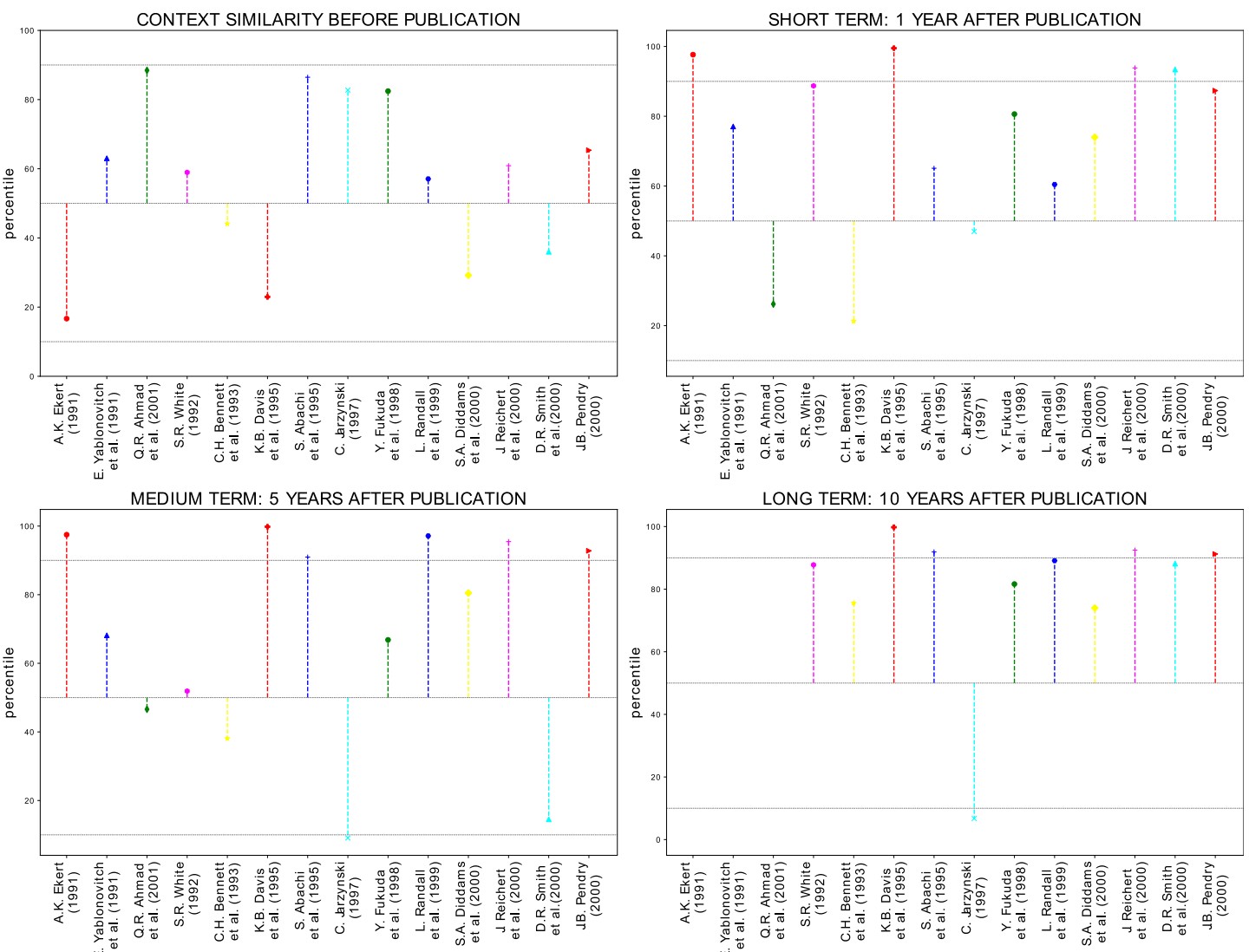

**Fig 4. Dynamics of the first two PACS of notable papers: On a longer time scale they seem to converge to a closer position.** The top-left panel shows the value of *context similarity* for PACS pairs of Nobel Prize winners and APS milestones calculated in the five years before their publication. The dashed lines represents, respectively, the 10th percentile, the median, and the 90th percentile of the *context similarity* distribution. The top-right panel and both bottom panels show the variation of *context similarity* of said pairs of PACS compared to the percentiles of the distribution of *context similarity* variations at different stages after the publication: one year, five years, and ten years in the future.

PACS through the years to spot a possible peculiar behavior of Nobels and milestones. In particular, we calculate such variations using as a starting point the value computed in the five years intervals having the publication year as the fifth, and last, year. The final value is computed at three different stages. The first one is set one year into the future after the date of publication, the second one five years into the future, and third one ten years into the future. Results are shown in Fig 4 together with the value of *context similarity* of all Nobel prize winners and milestones at the time of publication (top-left panel). In the remaining panels, each point corresponds to the variation of the PACS *context similarity* of these fundamental papers, while the horizontal lines correspond to median, the 10th percentile and the 90th percentile of distribution of the variation of all the other PACS pairs present in the same five-year period. In the short term, (top-right panel), there is a positive variation of *context similarity* for almost all

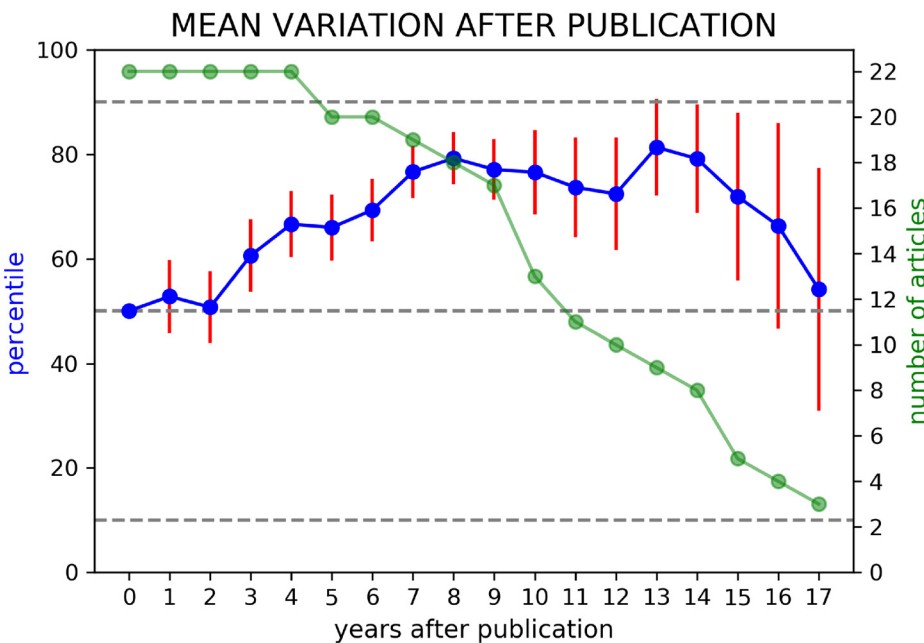

**Fig 5. We show an increase of the average similarity of the first two PACS of fundamental papers of Physics.** The trend of the mean value of the percentiles of all *context similarity* variations in Nobel prize papers and APS milestones, as a function of the year after publications (blue points), is positive up to several years after publication. Each point is represented with its standard error (vertical red lines). The highlighted percentiles (dashed grey lines), from top to bottom are 90th percentile, median, 10th percentile. The green line represents the number of papers available for the calculation of each blue point.

the articles. In the medium term (bottom-left panel) the situation is more mixed up: some articles experience a variation of *context similarity* which is outside the region delimited by the 10th and 90th percentile, while others experience an arrest. In the long term (bottom-right panel), we see that for almost all the Nobel prize winners and milestones, the variation tends to be at the tails of the distribution of *context similarity* variation. The conclusion we draw from Fig 4 is that the publication such as Milestones or Nobel, has a mixed impact on their PACS in the short and medium term, however, in the long term, with only one exception, they all experience an high increase. The fact that some pairs of PACS show negative trends for the variation of *context similarity* can be explained by them starting with high values at the time of publication that prevents the possibility to reach higher values. The interpretation we give to this situation is that such papers combine already strongly related PACS, while the others are pioneers in creating bridges between previously unrelated scientific areas.

In Fig 5, we show the average variation of *context similarity* of all these fundamental articles as a function of the number of years after their publication (red points). The error bars for each point represent the standard error relative to the average. Each point can be compared with the median and 10th and 90th percentiles of the variation of all the other PACS pairs of every article in the same time interval. The plot also shows the number of papers on which the average was carried out at each time (in green), which is decreasing with time due to the decrease of available papers on longer time spans. The plot shows that the *context similarity* undergoes a positive variation over the years after publication, which indicates that the main topics in these articles, identified by the first two PACS, are getting closer. This can be interpreted as an increase of interest in some fields of Physics related to the publication of those articles which have greatly influenced modern research. The negative trend in the last points

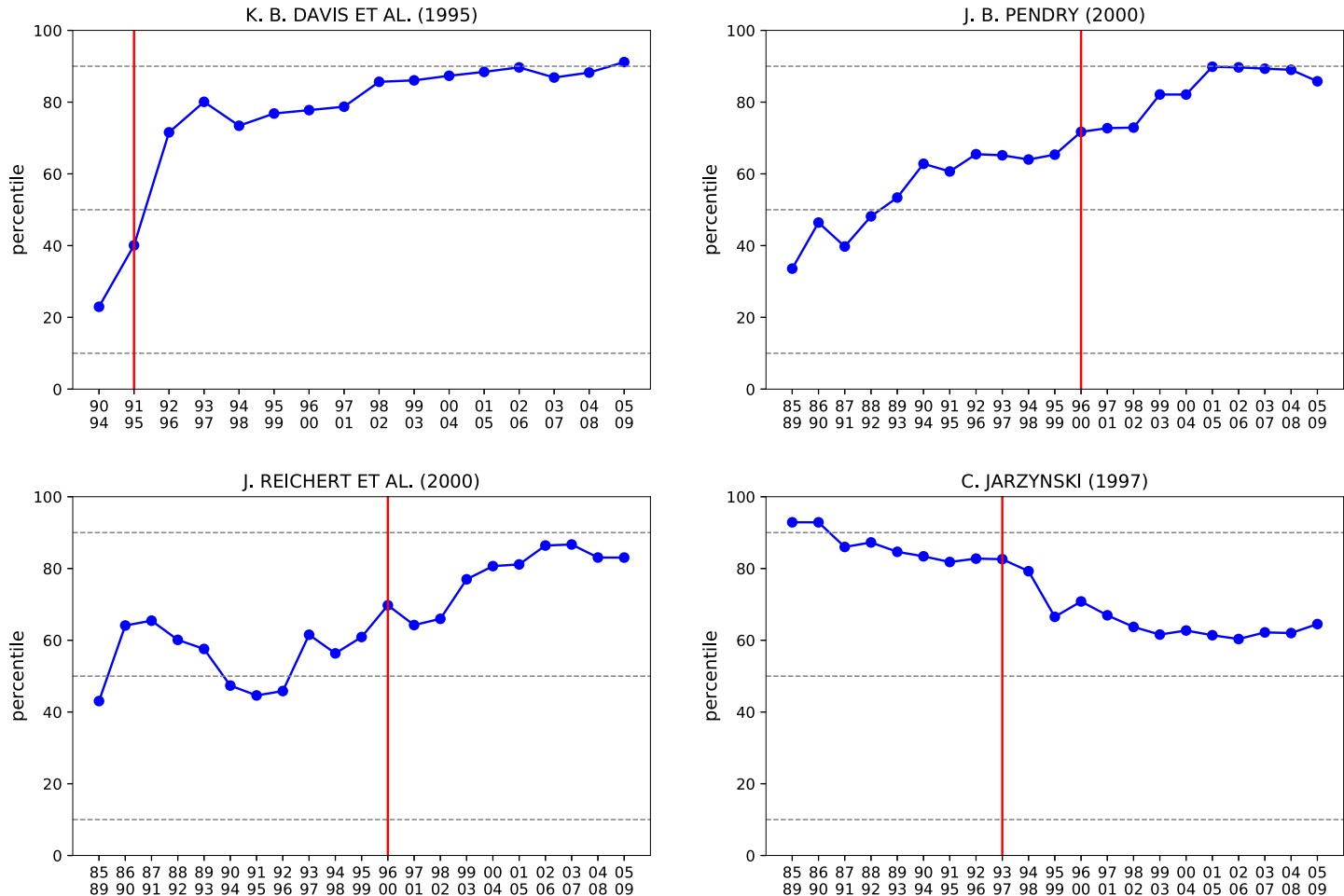

**Fig 6. We show the trend in time of the vaule of *context similarity* for the first two PACS of four fundamental papers in Physics expressed as percentile in the *context similarity* distribution.** The red vertical line indicates the publication year while the dashed gray line represents the 90th, the median, and the 10th percentile. In most cases, PACS with a value of *context similarity* close to the median of its distribution before the publication year, experience a steady growth. There are exceptions, such as Jarzynski (1997) [34], where the *context similarity* decreases after the publication, probably because the scientific interest moved towards different topics.

can be explained by noting that the value of the *context similarity* is now high enough not to undergo any further substantial changes. Moreover, in these points there is greater uncertainty of calculation due to the fact that the numbers of articles available is significantly reduced with respect to previous years.

Let us now focus on some specific examples. In Fig 6 we show the time evolution of *context similarity* of the first two PACS of four fundamental papers:

- **K.B. Davis et al. (1995)**: Bose-Einstein condensation in a gas of sodium atoms. [31]

- **J.B. Pendry (2000)**: Negative Refraction Makes a Perfect Lens. [32]

- **J. Reichert et al (2000)**: Phase Coherent Vacuum-Ultraviolet to Radio Frequency Comparison with a Mode-Locked Laser. [33]

- **C. Jarzynski (1997)**: Nonequilibrium Equality for Free Energy Differences. [34]

The vertical line represents the publication year.

We notice different behaviors: in three out of four examples, *context similarity* experiences a steady long-term growth after the publication. In the top right panel such growth is also anticipating the publication, a behavior similar to the one discussed in the previous section about the possibility to predict innovative combinations of scientific fields. In the bottom left panel, instead, *context similarity* decreases. As already discussed, we interpret these two different cases as the paper being either a pioneer in the field, thus paving the way to further research, or at the peak of research, from which it is only possible to climb down.

## 3 Conclusions

Describing and predicting the scientific progress is a challenging task. In this paper we use the APS database of physics articles to build a multi-dimensional space to investigate the relative motion of scientific fields, as defined by the PACS codes. Our machine learning methodology is based on Natural Language Processing techniques, which are able to extract the *context similarity* between words and, in our case, between scientific topics, starting from their presence in the APS articles. This vector representation permits to visualize in a clear way the trajectories in time of Physics topics and to predict innovations in Physics, as defined by the appearance of new combinations of PACS codes in APS articles. Finally, we observe that APS Milestones and Nobel winner papers have an effect in bringing together previously unrelated topics.

This work is a proof of concept that it is possible to go beyond standard network methodologies and build a space which is not only well suited to represent the dynamic of science, by it also allows to introduce metrics to make quantitative analysis and predictions. We believe that this research opens up a number of further developments, for example, this framework can be applied to study more extensive database, including not only Physics but also other scientific sectors and to investigate their mutual influence. Furthermore, it is an instrument that can be used to introduce more precise definition of scientific success such as one that links citations to the ability to affect the space of PACS: in future investigations for example, we plan to draw a comparison between sector's trajectories and the time evolution of citations.

## 4 Data and methods

### 4.1 Description of the data

The APS data-set (website: journals.aps.org/datasets) is a citation network data-set that is composed by papers in the field of physics organized by the American Physical Society. It contains 449935 papers in physics and related fields from 1977 to 2009. Among them, the high-impact papers used as evaluation benchmarks are derived from 78 milestone papers that experts from the American Physical Society have selected as outstanding contributions to the development of physics over the past 50 years. The PACS are alphanumerical strings hierarchically organized that are ascribed to scientific papers by authors at the time of publication and represent the domain of Physics the specific paper belongs to, for example the PACS $02.10.Yn$ indicates Matrix theory. The classification can be found in the supplementary information as a downloadable file.

### 4.2 Creation of pacs embeddings

PACS embeddings are created adapting the well-known algorithm of Word2Vec (in its Skip-Gram version) to our case of study [23, 36]. The code producing the results discussed is implemented in tensorflow [35], an open access deep learning library published by Google, that we adapted to process scientific papers and PACS. We refer to the literature for a detailed descriptions of the procedure behind Word2Vec [23, 36]. The key assumption is that there is a strong

parallel with Natural Language Processing: articles can be viewed as sentences, i.e. contexts, and PACS as words. Each PACS is initialized with a random vector (embedding) and the positions of such vectors are adjusted during the training in order to maximize the similarity between PACS belonging to the same context.

More into the details, each PACS is represented through a one-hot-encoded vector. This representation depends on the number of PACS to embed (the *vocabulary size*, in the language of [23, 36]): at 4 digits precision 500 PACS per training set. The one-hot-encoded vector corresponding to a PACS is a binary vector which has all zeros except for a single one in the position that the PACS under analysis occupies in the list of all PACS: the first code is represented by $[1, 0, 0, \ldots]$, the second code by $[0, 1, 0, \ldots]$ and so on. In this regard, a scientific paper is nothing else but a collection of PACS, i.e. a collection one-hot-encoded vectors.

To understand how Word2Vec works, we need to introduce two elements: an embedding matrix $E$ of size $V \times N$, where $V$ is the number of PACS to embed and $N$ the dimension of the embedding representation, a decoding matrix $D$ of size $N \times V$. Word2Vec is an iterative algorithm, at each steps a random batch of scientific papers is extracted from the training set and from each scientific paper in the batch, a random PACS is singled out as input and the remaining ones form the target context.

Let $h_i$ be the embedding vector of a given input PACS $p_i$. Let $P$ be the set of all the PACS $p_j$ forming the target context. The decoding matrix allows to calculate the score between the input PACS $p_i$ and all the words $p_j$ in the target context $P$. Let us call $u_{ji}$ the score for the j*th* PACS in the target context $P$, $u_{ji}$ is defined by:

$$u_{ji} = D_j \cdot h_i, \tag{1}$$

where $D_j$ is the j*th* column of the decoding matrix, obtained applying the the transposed matrix $D^T$ to the one-hot-encoded representation of the PACS $p_j$. Each score passes through a *softmax function* to become the posterior probability for the context PACS $p_j$ given the input PACS $p_i$:

$$\mathcal{P}(p_j | p_i) = \frac{\exp(u_{ji})}{\sum_{k=1}^{V} \exp(u_{ki})} \tag{2}$$

The posterior probability to predict the whole context given the input PACS is the product of all posterior probabilities for each PACS in the context.

$$\mathcal{P}(p_{j_1}, p_{j_2}, \cdots, p_{j_P} | p_i) = \prod_{j \in P} \mathcal{P}(p_j | p_i) \tag{3}$$

The Skip Gram model aims to maximize this probability at each step of the training for each input-context couple. However it is computationally more efficient to transform such maximization problem into the minimization of the following loss function:

$$L = -\log(\mathcal{P}(p_{j_1}, p_{j_2}, \cdots, p_{j_C} | p_i)) \tag{4}$$

At each step, Skip Gram is trained over a random batch of input-context couples therefore the total loss over the batch is the average of all the single losses $L$.

$$\mathcal{L} = \langle L \rangle \tag{5}$$

The training set is sampled in random batches at each training steps, this allows to efficiently process large quantities of data because parameter updates for Word2Vec are calculated only on subsets, i.e. only on those vectors present in the sample. At the end of the training, the

position of each codes mimics what the algorithm has learnt on the *scientific language* and allows to quantify the similarity between PACS given the global scientific production.

Word2Vec is trained through a variation of *Stochastic Gradient Descent*, therefore the embedding vectors will be different every time we run the algorithm, [37–39]. In particular, they can differ for two reasons: they can occupy different position in the space of PACS, and they can be randomly rotated with respect to the origin of the space in which they are defined. However, rotation invariant quantities, like the scalar product, can be still calculated and are not effected by rotations of the embeddings. We adopt the definition proposed by [21] of *context similarity* $CS_{ij}$ between PACS $i$ and $j$ as the average over 30 runs of the scalar product among the embeddings:

$$CS_{i,j} = \sum_{k}^{N_{run}} \frac{S_{ij}^k}{N_{run}}, \qquad (6)$$

where $S_{ij}^k$ is the scalar product between the embeddings of PACS $i$ and $j$ in the *kth* training instance. Taking the average over different runs offers two important advantages: on one hand, it allows us to check if the algorithm is learning to represent PACS correctly, by looking at the distribution of their scalar product, and on the other hand, it is a better proxy of the true *context similarity* between PACS.

The database at our disposal covers 25 years of scientific papers, from 1985 to 2009, we group them in 5-year-long overlapping intervals, from 1985-1989 up to 2005-2009, for a total of 21 time windows. We have empirically found that before 1985 there are not enough articles to make a statistically valid analysis. There are no a-priori instructions to identify the minimal size of the data-set required to have a good performance of the algorithm because this value depends on the database, the vocabulary size, and the aim of the training. As a general guideline, however, the documents used for training should make enough use of all the words in the vocabulary size. We have checked that if trained with too few papers, the embeddings of some of the PACS were randomly put in the space of PACS in multiple instances of the training. In other words, in order to create a reliable vector representation of PACS, the algorithm requires a sufficiently large training set, and this criterion is not met before 1985. This is due to the fact that before 1985 there are less than 2500 articles per year, while after 1985 this number jumps to 7500 and keeps growing to more than 15000 in 2009. Consequently, papers before 1985 are discarded and papers after 1985 are grouped in 5-years-long windows to have enough data in each sliding window to successfully train the model and produce reliable results. This choice is also theoretically motivated by the assumption that the time scale of the dynamics that shape the scientific research is longer than 5 years.

In each 5 years time interval we create a vector representations for the 500 more frequent 4-digit PACS. It has been empirically observed in [21, 22] that the algorithm is not able to create reliable vector representations for words that are too rare. We have verified that this number is a good compromise between having a wide spectrum of topics covered and the level of accuracy of the embeddings in terms of prediction power. This choice leaves out of out analysis around 40 PACS (with multiplicity less than 2) in the first sliding windows and around 150 PACS (with multiplicity less than 10) in the last sliding windows. The increase in the number of PACS left out and in their multiplicity is due to the positive trend in the number of published papers per year.

The embedding dimension chosen for this analysis is 8, i.e. PACS embeddings live in a 8-dimensional euclidean space. The optimal dimensionality, depending on the complexity of the problem under exam, and in particular on the size of the dataset and the vocabulary, is usually determined by a trial and error procedure, [23, 36], and our tests suggest that 8 is a good

compromise between efficiency and accuracy. The reader can find the code used to produce the embeddings at the following link: https://github.com/Andrea-Napoletano/WyFiG.

## Supporting information

**S1 Data.**
(ZIP)

**S2 Data.**
(ZIP)

**S3 Data.**
(ZIP)

**S4 Data.**
(ZIP)

**S5 Data.**
(ZIP)

**S6 Data.**
(ZIP)

**S7 Data.**
(ZIP)

**S8 Data.**
(ZIP)

**S9 Data.**
(ZIP)

**S10 Data.**
(ZIP)

**S11 Data.**
(ZIP)

**S12 Data.**
(ZIP)

**S13 Data.**
(ZIP)

**S14 Data.**
(ZIP)

**S15 Data.**
(ZIP)

**S16 Data.**
(ZIP)

**S17 Data.**
(ZIP)

**S18 Data.**
(ZIP)

**S19 Data.**
(ZIP)

**S20 Data.**
(ZIP)

**S21 Data.**
(ZIP)

**S22 Data.**
(ZIP)

**S23 Data.**
(ZIP)

**S24 Data.**
(ZIP)

**S25 Data.**
(ZIP)

**S26 Data.**
(ZIP)

**S27 Data.**
(ZIP)

**S28 Data.**
(ZIP)

**S29 Data.**
(ZIP)

**S30 Data.**
(ZIP)

**S31 Data.**
(ZIP)

**S32 Data.**
(ZIP)

**S1 File.**
(PDF)

**S2 File.**
(ZIP)

**S1 Fig.**
(HTML)

## Acknowledgments

The authors acknowledge Giulio Cimini for useful discussions. The authors acknowledge the anonymous referees for their comments that helped improve this paper.

## Author Contributions

**Conceptualization:** Andrea Napoletano, Andrea Zaccaria.

**Data curation:** Andrea Palmucci, Hao Liao.

**Formal analysis:** Andrea Palmucci, Andrea Napoletano.

**Funding acquisition:** Hao Liao.

**Investigation:** Andrea Palmucci.

**Methodology:** Andrea Zaccaria.

**Resources:** Hao Liao.

**Supervision:** Andrea Napoletano, Andrea Zaccaria.

**Validation:** Andrea Zaccaria.

**Writing – original draft:** Andrea Napoletano.

**Writing – review & editing:** Andrea Palmucci, Hao Liao, Andrea Napoletano, Andrea Zaccaria.

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
