## [Decision Letter · Decision Letter 0]

19 Feb 2020

PONE-D-19-32031

Where is your field going? A Machine Learning approach to study the relative motion of the domains of Physics

PLOS ONE

Dear Dr Napoletano,

Thank you for submitting your manuscript to PLOS ONE. After careful consideration, we feel that it has merit but does not fully meet PLOS ONE’s publication criteria as it currently stands. Therefore, we invite you to submit a revised version of the manuscript that addresses the points raised during the review process.

Both Reviewers agree that the manuscript is of high quality, but point out some issues especially regarding null model and presentation of results, which should be addressed in revision. They also offer some suggestions to improve the overall quality of the manuscript. 

We would appreciate receiving your revised manuscript by Apr 04 2020 11:59PM. To enhance the reproducibility of your results, we recommend that if applicable you deposit your laboratory protocols in protocols.io, where a protocol can be assigned its own identifier (DOI) such that it can be cited independently in the future. For instructions see: http://journals.plos.org/plosone/s/submission-guidelines#loc-laboratory-protocols

We look forward to receiving your revised manuscript.

Kind regards,

Roberta Sinatra

Academic Editor

PLOS ONE

Journal Requirements:

Reviewers' comments:

Reviewer's Responses to Questions

**Comments to the Author**

1. Is the manuscript technically sound, and do the data support the conclusions?

Reviewer #1: Yes

Reviewer #2: Yes

2. Has the statistical analysis been performed appropriately and rigorously? 

Reviewer #1: Yes

Reviewer #2: Yes

3. Have the authors made all data underlying the findings in their manuscript fully available?

Reviewer #1: Yes

Reviewer #2: Yes

4. Is the manuscript presented in an intelligible fashion and written in standard English?

Reviewer #1: Yes

Reviewer #2: Yes

5. Review Comments to the Author

Reviewer #1: The paper is well-written and original, and I enjoyed reading it. There are very few minor areas of improvement that I will suggest, but I do not consider them a requirement for publication; the paper is good enough as it is and I only put these forward in the hope of making it stronger. To be clear: if you agree with my suggestions do implement them, but do not consider them binding. The suggestions are not in order of importance.

a) Figure 1 does not print well in black and white. You might consider using colors that print very differently in black and white for adjacent clusters, to increase contrast.

b) Most readers that are familiar with Word2Vec are probably familiar with t-SNE, but Dynamic t-SNE is less known. A few lines in the Methods to explain the difference between t-SNE and Dynamic t-SNE might make the paper more readable. (A detailed explanation is not necessary, since you cited the appropriate papers).

c) Lines 139-140: I am not sure that it is common knowledge that AUC ROC is always 0.5 for random guessing, even in your case (i.e. the presence of unbalanced classes). Maybe mention it to increase readability?

d) Lines 142-143. I would not use the wording "a reasonable high number of positive(s) have been dentified", since there is no alternative baseline model to compare the F-score to. I appreciate the novelty of the method, which is a proof of concept, and this means you do not have anything else to compare it to. So I think a baseline other than the random model you already included is not necessary.

e) Lines 174-176. Minor suggestion (since these results are not the main point of the paper). You are open to the objection that, depending on what the distribution of context similarity variation over time is, using units of standard deviation does not prove statistical significance. Since you have the full distribution of variations available and enough computing power, why not use the percentiles of that distribution instead of its standard deviation? It would allow to compute the p-value and therefore gauge the significance immediately. With this method you could also use a Bonferroni test to numerically prove that the results in Figure 4 (and lines 183-186) are statistically significant. Alternatively, you could say more about the properties of the distribution of context similarity variation, and argue that the standard deviation is a good metric to use in this case.

f) Lines 324-236: You suggest that there is a threshold amount of data below which the algorithm is not reliable. What do you estimate this threshold to be? If you have a good argument, how do you estimate the threshold? These considerations could be very useful in the interest of reproducibility and for people who want to use Word2Vec in new contexts.

g) I recommend trying to publish the code for your analysis open-source, in the interest of reproducibility.

Again, I liked the paper very much. I gave a "minor revision" suggestion so that you get a chance to fix the typos. I am then inclined to accept the manuscript independently of whether you implement any of the suggestions above. I look forward to reading your future works on the topic and I hope my notes were useful to you.

---

English language suggestions for improvement:

a) Lines 131-138 could use some clarification. Especially point 2 (line 136): "We classify the set couples of PACS in two separate classes according to their possible co-occurrence." I had to re-read it a couple of times and use context to understand that the two classes are "couples that exist in the test set" and "couples that do not exist in the test set". Why not write it explicitly?

b) Lines 256-259: maybe add a couple of PACS as an example to let the reader understand what they are. (I appreciate that there are many examples throughout the text, but I went to check this section, to understand what a PACS was, before encountering any example).

c) Line 266: Very minor style note. I would use "published" instead of "realized".

Typos:

d) line 308: 'Stochastic' instead of the typo 'stocastic'.

e) Lines 142-143. "a reasonable high number of positive". It should be "positives", plural, and "reasonably", adverb.

Reviewer #2: The paper introduces a methodology that uses NLP techniques to track the similarity between research topics within physics and its dynamics over time. Moreover, it shows the effectiveness of this methodology in mapping the landscape of research topics, forecast their combination and estimate the impact of milestones. The methodology is scientifically sound, the paper is clear and well written and results are interesting and compelling, thus I certainly advocate its acceptance. In what follows I list some minor comments that the authors may consider before proceeding with the publication, sorted by order of importance:

- In section 2.2, the authors use the new introduced context similarity to predict the appearance of new couples of PACS. I understand their method and results, but I think it would be interesting to discount their findings with a null model that takes into account the different growth of fields. Indeed, the random guess to which the authors compare their results consider all PACS as equal sized: given a set of fields, all possible combinations are considered equivalent. In fact, as a result of the well known inflation of science, the size of research topics labelled with PACS is growing in time and, in general, different fields can grow at different rates. For this I suggest the implementation of a very basic null model that compute the probability of having a new combination of PACS as the result of their relative growth with respect to the rest of the fields.

- Among references [15-19] I suggest adding Gerlach et al. "A network approach to topic models", Science advances, 2018 as an alternative to topic modeling based on NLP

- In Fig. 1, it would be interesting to highlight the starting and ending point of the embedded dynamics of each PACS to know the temporal direction of their evolution. This can be made simply by using differently colored dots to pinpoint start and end of each trajectory.

- At page 7, line 209 the sentence should read "... the fact that the NUMBER OF articles available is..."

6. PLOS authors have the option to publish the peer review history of their article (what does this mean?). If published, this will include your full peer review and any attached files.

Reviewer #1: No

Reviewer #2: No

---

## [Author Response · Author response to Decision Letter 0]

15 May 2020

Dear editor,

first of all we would like to thank the referees for their comments, that gave us the possibility to make our paper more readable and more scientifically robust. We have incorporated practically all suggestions in the revised version of our manuscript, and we believe that these resulted in a much improved work.

Please find our point-by-point response below. 

Best regards

the authors

Reviewer 1:

a)Figure 1 does not print well in black and white. You might consider using colors that print very differently in black and white for adjacent clusters, to increase contrast.

We have tried different combination of colors to achieve this goal, however, we found that a good image in black and white we had to use too many light colors, which were difficult to read in the colored version. For this reason, we have decided to keep the original figure. You can find some of our attempts at the end of this letter.

b) Most readers that are familiar with Word2Vec are probably familiar with t-SNE, but Dynamic t-SNE is less known. A few lines in the Methods to explain the difference between t-SNE and Dynamic t-SNE might make the paper more readable. (A detailed explanation is not necessary, since you cited the appropriate papers).

We ha added two sentences to clarify this point (lines 83-87 of the revised version).

c) Lines 139-140: I am not sure that it is common knowledge that AUC ROC is always 0.5 for random guessing, even in your case (i.e. the presence of unbalanced classes). Maybe mention it to increase readability?

We have better clarified this point (lines 167-169) and added, as explicitly requested by reviewer 2, a null model to control our results with respect to PACS of different and time dependent sizes. 

d) Lines 142-143. I would not use the wording "a reasonable high number of positive(s) have been dentified", since there is no alternative baseline model to compare the F-score to. I appreciate the novelty of the method, which is a proof of concept, and this means you do not have anything else to compare it to. So I think a baseline other than the random model you already included is not necessary.

We agree, and we have removed this sentence.

e) Lines 174-176. Minor suggestion (since these results are not the main point of the paper). You are open to the objection that, depending on what the distribution of context similarity variation over time is, using units of standard deviation does not prove statistical significance. Since you have the full distribution of variations available and enough computing power, why not use the percentiles of that distribution instead of its standard deviation? It would allow to compute the p-value and therefore gauge the significance immediately. With this method you could also use a Bonferroni test to numerically prove that the results in Figure 4 (and lines 183-186) are statistically significant. Alternatively, you could say more about the properties of the distribution of context similarity variation, and argue that the standard deviation is a good metric to use in this case.

We thank the referee for pointing out this. We have checked that the distribution of context similarity variation does not follow a gaussian distribution. For this reason, we have changed the figures 4,5, and 6 to show the percentiles of the distribution that, as suggested, represent a more appropriate statistical benchmark. We point out that, however, the main scientific conclusions are not affected. 

f) Lines 324-236: You suggest that there is a threshold amount of data below which the algorithm is not reliable. What do you estimate this threshold to be? If you have a good argument, how do you estimate the threshold? These considerations could be very useful in the interest of reproducibility and for people who want to use Word2Vec in new contexts.

Unfortunately, there is no clear-cut recipe to estimate the threshold as a function of the various data features and the final aim of the training. We have added a short paragraph to comment this point (lines 367-373).

g) I recommend trying to publish the code for your analysis open-source, in the interest of reproducibility.

We fully agree on this point. We have prepared a github with our codes and a data sample to reproduce our results: https://github.com/Andrea-Napoletano/WyFiG.

Again, I liked the paper very much. I gave a "minor revision" suggestion so that you get a chance to fix the typos. I am then inclined to accept the manuscript independently of whether you implement any of the suggestions above. I look forward to reading your future works on the topic and I hope my notes were useful to you.

Thank you very much for your interest in our work and for your suggestions, that we believe improved the paper a lot.

---

English language suggestions for improvement:

a) Lines 131-138 could use some clarification. Especially point 2 (line 136): "We classify the set couples of PACS in two separate classes according to their possible co-occurrence." I had to re-read it a couple of times and use context to understand that the two classes are "couples that exist in the test set" and "couples that do not exist in the test set". Why not write it explicitly?

We have clarified this point (see lines 141-144)

b) Lines 256-259: maybe add a couple of PACS as an example to let the reader understand what they are. (I appreciate that there are many examples throughout the text, but I went to check this section, to understand what a PACS was, before encountering any example).

We have added an example (see line 284). We have also provided the list of PACS as supplementary information and corrected a reference in the text since the website originally referenced is no longer available.

c) Line 266: Very minor style note. I would use "published" instead of "realized".

We have corrected the text accordingly.

Typos:

d) line 308: 'Stochastic' instead of the typo 'stocastic'.

We have corrected the typo

e) Lines 142-143. "a reasonable high number of positive". It should be "positives", plural, and "reasonably", adverb.

We have removed the sentence

Reviewer #2: 

The paper introduces a methodology that uses NLP techniques to track the similarity between research topics within physics and its dynamics over time. Moreover, it shows the effectiveness of this methodology in mapping the landscape of research topics, forecast their combination and estimate the impact of milestones. The methodology is scientifically sound, the paper is clear and well written and results are interesting and compelling, thus I certainly advocate its acceptance. In what follows I list some minor comments that the authors may consider before proceeding with the publication, sorted by order of importance:

- In section 2.2, the authors use the new introduced context similarity to predict the appearance of new couples of PACS. I understand their method and results, but I think it would be interesting to discount their findings with a null model that takes into account the different growth of fields. Indeed, the random guess to which the authors compare their results consider all PACS as equal sized: given a set of fields, all possible combinations are considered equivalent. In fact, as a result of the well known inflation of science, the size of research topics labelled with PACS is growing in time and, in general, different fields can grow at different rates. For this I suggest the implementation of a very basic null model that compute the probability of having a new combination of PACS as the result of their relative growth with respect to the rest of the fields.

We thank the referee for this suggestion, that gives an important improvement of our results in terms of robustness. Following her/his suggestions, we have implemented a null model that takes into account the relative growth of each field. Using the curveball algorithm (ref. 27) we randomized the articles-PACS network without changing the degrees of neither the papers nor the PACS, and so preserving the time evolution of their respective sizes. We modified figure 2 accordingly. The performance of this null model is above a random guess (AUC 0.5), but it is still outperformed by the context similarity.

- Among references [15-19] I suggest adding Gerlach et al. "A network approach to topic models", Science advances, 2018 as an alternative to topic modeling based on NLP

We have added the suggested reference and a short sentence to comment it.

- In Fig. 1, it would be interesting to highlight the starting and ending point of the embedded dynamics of each PACS to know the temporal direction of their evolution. This can be made simply by using differently colored dots to pinpoint start and end of each trajectory.

We agree with this request, however, trying to highlight the start and end point of each trajectory we have observed that the figure loses its readability because of the high number of trajectories: adding such a high number of start or end points on the plot creates overlaps among trajectories and this good intent results in a very confused plot. For this reason, we decided to leave the original figure in the paper, while uploading as supporting material an interactive plot in .html format that address this point.

- At page 7, line 209 the sentence should read "... the fact that the NUMBER OF articles available is..."

We have corrected the typo

---

## [Editor Report · Decision Letter 1]

18 May 2020

Where is your field going? A Machine Learning approach to study the relative motion of the domains of Physics

PONE-D-19-32031R1

Dear Dr. Napoletano,

We are pleased to inform you that your manuscript has been judged scientifically suitable for publication and will be formally accepted for publication once it complies with all outstanding technical requirements.

With kind regards,

Roberta Sinatra

Academic Editor

PLOS ONE

Additional Editor Comments (optional):

The authors have thoroughly and successfully addressed all the minor comments raised by the Reviewers. I recommend the paper for publication.
---

## [Editor Report · Acceptance letter]

22 May 2020

PONE-D-19-32031R1 

Where is your field going? A Machine Learning approach to study the relative motion of the domains of Physics 

Dear Dr. Napoletano:

I am pleased to inform you that your manuscript has been deemed suitable for publication in PLOS ONE. Congratulations! Your manuscript is now with our production department. 

With kind regards,

on behalf of

Prof. Roberta Sinatra 

Academic Editor

PLOS ONE